# When does nitrate peak in rivers and why? Catchment traits and climate drive synchrony with discharge

Lu Yang<sup>1</sup>, Kieran Khamis<sup>1</sup>, Julia L.A. Knapp<sup>2</sup>, Joshua R. Larsen<sup>1,3</sup>

<sup>1</sup>School of Geography, Earth and Environmental Sciences, University of Birmingham, Birmingham, B15 2TT, United Kingdom

<sup>2</sup>Department of Earth Sciences, Durham University, Durham, DH1 3LE, United Kingdom

<sup>3</sup>Birmingham Institute for Forest Research (BIFoR), University of Birmingham, B15 2TT, United Kingdom

Correspondence to: Lu Yang (l.yang.7@pgr.bham.ac.uk or im.yanglu@hotmail.com)

Abstract. Anthropogenic nitrogen loading has disrupted global biogeochemical cycles, degrading water quality and altering ecosystem functions. Rivers mediate nitrogen transport and reactivity, yet at the seasonal scale, the temporal links between peak river nitrate concentrations (N) and water flow (Q) are poorly understood. Here, we used the approach of Weighted Regressions on Time, Discharge, and Season (WRTDS) to reconstruct daily timeseries of N concentrations from routine monitoring data. These were used to assess the long term N-Q synchrony and its variability across 66 river catchments in England (2000-2019), and a Random Forest Model was used to identify the key controls on each synchrony type. This revealed three general behaviours: 1) smaller catchments dominated by agriculture displayed peak N during high flow (QMax-Synced, 28.8% of catchments), 2) larger and/or more urbanised catchments had the highest N concentrations during low flow periods likely due to point-source inputs (OMin-Synced, 25.8% of catchments), and, 3) larger highly mixed land use catchments displayed a decoupling of N and flow conditions, i.e. were asynchronous (Asynced, 46.8% of catchments). The temporal consistency of peak N-Q synchrony was determined by the dominant hydrological processes and their interaction with anthropogenic pressures. In QMax-synced catchments, wetter winters, and steeper slopes promoted more rapid runoff, reinforcing synchrony. In QMin-synced catchments, synchrony reflected the dominance of urban point-source inputs (represented as urban area and population density) but was sustained only under sufficiently extreme low flows. Asynced catchments showed the greatest year-to-year switching in the dominant synchrony year type, with wetter years likely enhanced groundwater recharge and legacy-N delivery, favouring QMin-like behaviour, whereas years dominated by rapid runoff and shallow flow paths promoted QMax-like winter flushing. Our findings reveal that nitrate-discharge synchrony is not fixed but dynamically regulated by hydroclimatic variability, catchment connectivity, and human infrastructure. Framing nitrate export through synchrony exposes a critical temporal dimension of nutrient cycling that a purely spatial analyses of loads or concentrations would overlook, providing new insight into how climatic and anthropogenic forcing interact to shape water-quality responses in human-modified landscapes.

20

25

10

#### 1 Introduction





River networks are critical conduits in the global nitrogen (N) cycle, transporting and transforming nitrogen from terrestrial landscapes to lakes and coasts. However, intensive anthropogenic activities, such as fertilizer application, wastewater discharge, and land drainage, have disrupted this natural nitrogen cycling, leading to widespread water quality degradation and ecological damage (Galloway et al., 2008; Zhang et al., 2015; Diaz and Rosenberg, 2008). While much attention has been given to the magnitude of nitrogen loads, the seasonal dynamics of nitrate delivery and their alignment with hydrological processes remain poorly understood, particularly at broader spatial scales.

River discharge is a dominant control on nitrogen export, particularly in agricultural regions where flow variability can explain up to 75–93% of annual nutrient fluxes (Ezzati et al., 2022). Yet, the timing of peak nitrate concentrations relative to peak or minimum flows remains poorly understood, especially at the interannual scale. These peaks often dominate annual loads and drive the most acute ecological impacts as well as regulatory exceedances. While concentration—discharge (C—Q) relationships offer valuable insights into source—mobilisation dynamics and hydrological controls (Musolff et al., 2015; Bieroza et al., 2018; Knapp and Musolff, 2024), interpreted in isolation they risk emphasising average behaviour and often obscure temporal shifts in nitrate delivery pathways.

The analysis of synchrony, broadly defined as the temporal alignment of two or more processes, was initially developed in ecology to evaluate and interpret ecosystem attributes and processes such as trophic interactions, metapopulation dynamics (Bjørnstad et al., 1999; Hanski, 1998). To better understand interannual hydrological controls on peak nitrate concentrations and their timing, we apply the concept of synchrony, which we describe as the degree to which seasonal nitrate peaks align with seasonal patterns in flow, such as the month of highest or lowest discharge. Thus, synchrony captures the seasonal alignment (or misalignment) of biogeochemical responses with hydrological forcing, offering a dynamic lens on catchment function (Van Meter et al., 2019; Abbott et al., 2018). Despite its potential, few studies have quantified nitrate—discharge synchrony over time, and even fewer have explored how synchrony varies across diverse landscapes and land use types. This peak-based, month-focused metric of synchrony offers a simple categorical classification of catchments, providing insight into management-critical periods. By focusing on the timing of annual peaks, it is also less sensitive to short-term fluctuations and data gaps. Meanwhile, the C-Q slope, quantifying the magnitude and direction of nitrate response to hydrological variation, are often used to explain the spatial distribution and availability of solutes within catchments (Zhi and Li, 2020; Knapp et al., 2022). Combining the synchrony metric and C-Q slope analysis thus clarifies both when seasonal extremes occur and how nitrate is delivered under varying hydrological conditions.

This study addresses our limited understanding of N-Q synchrony, and its variation across different hydrological and anthropogenic conditions, by examining the timing of nitrate concentration peaks relative to seasonal flow patterns in 66 catchments across England. Using long-term datasets and reconstructed concentration time series, we aim to identify distinct synchrony modes and investigate their drivers. The specific objectives of this study are to: 1) identify and characterise catchments based on their dominant synchrony patterns between discharge and nitrate concentration, 2) determine how

catchment properties such as baseflow, slope, and wastewater infrastructure shape synchrony behaviour, and 3) assess how interannual climate variability and anthropogenic pressures drive synchrony shifts, particularly in catchments with mixed or competing nitrate sources.

By identifying dominant synchrony behaviours, we seek to reveal how hydrological regimes, land use, and legacy pollution interact to control seasonal nitrate export, with implications for targeted nutrient management under changing climate and land use pressures.

#### 0 2 Data and methods



the catchment area.

#### 2.1 Data sources and screening

Water quality data for site across England were obtained from the UK Environment Agency's Water Quality Archive (EA, 2020). The dataset consisted of river water quality measurements collected at irregular intervals from 2000 to 2020. We focused on nitrate-N concentrations as the dominant form of dissolved N (NO<sub>3</sub>-N). Initial screening of the data identified 21,049 sampling sites with NO<sub>3</sub>-N data. Daily mean discharge records were obtained from the National River Flow Archive (NRFA) which contains daily discharge measurements for over 1500 UK gauging stations.

We applied the following criteria to filter and select nitrate data for our study: (1) the data needed to cover at least 80% of the months between 2000 and 2019, (2) the time series could not include gaps longer than 3 consecutive months, (3) each water quality site must be within 1 km of a flow gauge with daily discharge (Q) data and located on the same river, and; (4) the flow gauge must have at least 90% of the discharge data available over the 20-year period. Application of the criteria above resulted in 66 catchments (Fig. S1) with nitrate and corresponding discharge time series data. In total, there were 18,947 nitrate concentration observations and 482,130 flow measurements across these 66 sites and the 20 years of study. All data analyses were based on the water year (from 1st October to 30th September of the subsequent year).

The selected catchments spanned a wide range of hydrological, topographical, land cover, geological, lithology and soil characteristics and other descriptors (Table 1, (NRFA, 2020). The Standardized Precipitation Index (SPI) was obtained for 56 out of 66 studied catchments from the UK Centre for Ecology & Hydrology (UKCEH), via the UK Water Resources Portal (UKCEH, 2024). The standard precipitation index (SPI) expresses precipitation anomalies in units of standard deviation relative to a long-term baseline (Mckee et al., 1993). We adopted the UK gridded SPI data set of (Tanguy et al., 2017), which used 1961–2010 as its reference period. The SPI can be calculated for different accumulation periods: for example, SPI1 reflects precipitation anomalies over a 1-month window, while SPI12 reflects anomalies accumulated over 12 months. For each catchment, we defined winter SPI1 as the mean of the monthly SPI1 values for December to February, and annual SPI12 as the September SPI12 value for each water year. The population density was extracted and calculated from UK gridded population 2011 based on Census 2011 published by Environmental Information Data Centre (Reis et al., 2017). The density of Wastewater Treatment Plant (WWTPs) was calculated by dividing the numbers of WWTPs in a catchment by

3

Table 1. Key catchment properties calculated and used in subsequent analyses. To aid understanding a more detailed description of each variable is provided.

| Category          | Variable                           | Unit            | Description (NRFA,2020)                       |  |  |
|-------------------|------------------------------------|-----------------|-----------------------------------------------|--|--|
| Topography        | Catchment Area                     | km <sup>2</sup> | Area of the catchment at gauging location     |  |  |
|                   | Mean Altitude                      | m               | Mean catchment altitude                       |  |  |
| Lithology & Soils | ogy & Soils High Permeable Bedrock |                 | Proportion of highly permeable bedrock        |  |  |
|                   | Moderate Permeable                 | %               | Proportion of moderately permeable bedrock    |  |  |
|                   | Bedrock                            |                 |                                               |  |  |
|                   | Low Permeable Bedrock              | %               | Proportion of low permeability bedrock        |  |  |
|                   | High Permeable Surface             | %               | Proportion of High-permeability surface       |  |  |
|                   |                                    |                 | deposits                                      |  |  |
|                   | Low Permeable Surface              | %               | Proportion of Low-permeability surface        |  |  |
|                   |                                    |                 | deposits                                      |  |  |
| Land Cover        | Wood Land                          | %               | Percentage of Woodland cover                  |  |  |
|                   | Arable Land                        | %               | Percentage of Arable and horticultural land   |  |  |
|                   | Grass Land                         | %               | Percentage of Grassland cover                 |  |  |
|                   | Mountain Heath Bog                 | %               | Percentage of Mountain heath and bog cover    |  |  |
|                   | Urban Land                         | %               | Percentage of Urban land cover                |  |  |
| Hydrology         | BFI                                | -               | Baseflow Index                                |  |  |
|                   | PROPWET                            | %               | The Proportion of Time Soils Are Wet          |  |  |
|                   | FARL                               | -               | The Flood Attenuation by Reservoirs and       |  |  |
|                   |                                    |                 | Lakes index                                   |  |  |
|                   | SPR                                | %               | Standard Percentage Runoff Coefficient: the   |  |  |
|                   |                                    |                 | percentage of rainfall typically converted to |  |  |
|                   |                                    |                 | surface runoff                                |  |  |
|                   | DPS                                | m/km            | Mean Drainage Path Slope                      |  |  |
| Climate           | Winter SPI1                        | -               | Mean Monthly SPI1 value of the winter months  |  |  |
|                   |                                    |                 |                                               |  |  |
|                   | SPI12                              | -               | Annual SPI12 value for September of each      |  |  |
|                   |                                    |                 | water year                                    |  |  |
| Anthropogenic     | Population Density                 | Persons/        | Number of people per square km                |  |  |
|                   |                                    | $\mathrm{km}^2$ |                                               |  |  |
|                   | WWTPs density                      | no./km²         | Number of WWTPs per square km                 |  |  |






# 2.2 Modelling daily data and C-Q definitions

Daily concentrations were reconstructed using Weighted Regression on Time, Discharge and Season (WRTDS), which was implemented in the R package EGRET (version 3.0.9) (Hirsch, 2023; Hirsch et al., 2010). This approach estimates daily concentrations from irregularly sampled data using a locally weighted regression:

$$\ln (C_i) = \beta_{0,i} + \beta_{1,i}t_i + \beta_{2,i} \ln (Q_i) + \beta_{3,i} \sin (2\pi t_i) + \beta_{4,i} \cos(2\pi t_i) + \varepsilon_i$$

where  $t_i$  is the time in decimal years,  $C_i$  is the concentration on day i,  $Q_i$  is the daily discharge,  $\beta_0$  is the intercept,  $\beta_1$  captures long-term concentration trends,  $\beta_2$  describes the sensitivity of a change in concentration to a change in discharge (i.e. the C-Q slope), and  $\beta_3$  and  $\beta_4$  account for seasonal cycles, and  $\varepsilon_i$  is the residual error term. These coefficients are fitted through regression at each time point, weighting observations by similarity in time, discharge, and season. This dynamic approach allows the concentration-discharge relationship to vary smoothly over time, reduces bias from irregular sampling (especially under-representation of high flows), and handles censored values effectively (Hirsch and De Cicco, 2015). We further used the locally estimated  $\beta_2$  coefficients to classify each catchments export regime as dilution ( $\beta_2$ <0.1), chemostasis (-0.1 $\leq \beta_2 \leq$ 0.1), and mobilisation ( $\beta_2$ >0.1) following established thresholds (Zhang, 2018; Herndon et al., 2015). The R scripts published by (Zhang et al., 2016) were used to estimate and extract the  $\beta_2$  coefficients. To support seasonal synchrony analysis, we aggregated the reconstructed daily concentrations This produced a more complete and temporally consistent representation of nitrate dynamics than the observed data alone and enabled robust comparisons across catchments and years (comparisons of measured and fitted concentrations are shown in Fig. S2-S7).

## 2.3 Defining Synchrony

We first confirmed that both discharge and nitrate concentration exhibited a distinct unimodal pattern at all sites. We then identified, for each site and each year, the months of maximum and minimum discharge, as well as the month of the maximum nitrate concentration. Based on the consistency between these seasonal timings, we first defined the annual synchrony status for each year at each site as: 1) *QMax-Synced* when the maximum concentration month coincided with or fell within one month (±1 month) of the maximum discharge month, 2) *QMin-Synced* using the same criteria as for QMax-Synced, except applied to the minimum discharge month, or 3) *Asynced* for years/catchments that met the criteria for neither category. This one-month coincidence window was chosen because it corresponds to the temporal resolution of seasonal flow regimes in temperate catchments. A narrower window would risk fragmenting the seasonal synchrony into noise, whereas a broader one would blur seasonality into semi-annual behaviour. We then used these annual synchrony categories in two ways throughout the analysis. First, as a catchment level classification, where each catchment was assigned a dominant category (QMax-synced, QMin-synced) based on the majority (>50%) of its annual classifications while all others were grouped as the Asynced. Second, we assessed the consistency of synchrony within each catchment across years, using the proportion of years in which it exhibited QMax- or QMin-synchrony as a measure of interannual variability. This dual







approach allowed us to characterise both the dominant synchrony pattern at each site and its temporal stability (or lack of) over two decades of record.

#### 2.4 Statistical analysis with catchment characteristics

To analyse the spatial variability of nitrate concentrations in rivers, the mean discharge and nitrate concentration for each catchment were calculated. The ratio of the coefficients of variation (CV) of discharge (CVq) and concentrations (CVc) was also calculated for each site across years to understand the hydrological impact on nitrate concentrations in each catchment. To understand the catchment controls on the two synchrony patterns, a series of catchment descriptors (Table 1) were selected for the next step, where Random Forest (RF) modelling was applied to examine the relationship between the spatial patterns in synchrony and catchment characteristics. RF analysis was chosen for its robustness and ability to handle complex interactions within the data (Breiman, 2001). It is a non-parametric machine learning model composed of multiple decision trees, which builds each decision tree by randomly selecting attributes and optimally splitting the data. In our analysis, a random forest classification model was trained for the two significant synchrony patterns (QMax-Synced and QMin-Synced) to recognise the factors controlling the different synchrony patterns. Asynced Catchments were excluded as this group represents catchments with no dominant synchrony pattern and high variability, which would reduce the clarity and interpretability of the results. A Random Forest classifier implemented via the efficient and scalable ranger package in R was chosen for its robustness to small sample sizes and its ability to handle complex, nonlinear relationships between features (Wright and Ziegler, 2017). A three repeated 10-fold cross-validation approach was employed to evaluate model performance and ensure generalizability. The model was configured to compute permutation-based variable importance, allowing identification of the most influential descriptors for classification (p

QMax- and non-QMax years in catchments where QMax-synchrony was more frequent and compared MinQ percentiles between QMin- and non-QMin years in catchments where QMin-synchrony was more frequent. Last, we used Spearman rank correlations and non-parametric Wilcoxon rank-sum tests to assess the relationship between synchrony variability and potential influential factors, calculated and visualised using the numpy, pandas, seaborn, matplotlib and scipy packages in Python (Virtanen et al., 2020; Reback et al., 2020; Harris et al., 2020; Waskom, 2021; Hunter, 2007).

#### 3 Results


To ensure the reliability of modelled nitrate concentrations, we assessed the goodness-of-fit of the WRTDS models for each catchment. Across all sites, the median R<sup>2</sup> value was 0.59 ranging from 0.11 to 0.82, and the median RMSE was 0.60 mg/L, ranging from 0.20 to 4.12 mg/L. Monthly comparisons of observed and modelled concentrations for each site are provided in the Supplementary Information (Figures S2–S7).

## 3.1 Spatial patterns across and temporal variability within catchments

Catchment synchrony classifications and their temporal stability differed across the 66 catchments (Fig. 1 & 2). QMax-synced catchments (28.8%) were primarily located in southern and southwestern England, where nitrate peaks typically aligned with winter high flows. QMin-synced catchments (25.8%) were concentrated in the northwest, showing peak nitrate concentrations during or near summer low flows. The Asynced catchments (46.8%) were more broadly distributed across the country and did not exhibit a consistent seasonal alignment between nitrate and discharge.

Figure 1: Spatial Distribution of Synchrony Patterns across England. Sites classified as QMax-Synced show consistent alignment between peak nitrate (N) and discharge (Q) timing (± 1 month) in over 50% years. QMin-Synced sites represent those where over 50% of years align with minimum discharge months. Asynced sites represent all other cases.



Figure 2: Interannual Consistency of Synchrony Patterns. The QMax-Synced segments represent the percentage of years where the peak nitrate concentration and peak discharge occur around the same time of year (±1 month). The QMin-Synced segments indicate the percentage of years where the peak nitrate concentration coincides with the time of minimum discharge (±1 month). The Asynced segments represent the percentage of years in between (i.e., neither QMax- nor QMin synced years).

QMax-synced sites exhibited clear winter nitrate peaks (Fig. 3a), with highest median concentrations in February (6.45 mg L<sup>-1</sup>, IQR: 4.19 – 6.95 mg/L) and lowest in September (4.39 mg L<sup>-1</sup>, IQR: 2.80 – 5.70 mg/L). In contrast, QMinsynced catchments typically showed a reversed seasonal pattern, with lowest concentrations during winter high flow (January: 4.40 mg/L, IQR: 3.12–4.95 mg/L) and peak concentrations during summer low flow (July: 6.15 mg/L, IQR: 4.15–8.02 mg/L; Fig. 3b). Asynced sites displayed flatter, more spatially variable nitrate regimes, without a dominant seasonal signal. While individual sites may have distinct seasonal patterns, the lack of alignment in their timing and magnitude resulted in the group-average curve appearing flat. Despite these marked seasonal differences, median nitrate concentrations did not differ significantly among the three synchrony types (Fig. S8). Overall, interannual variability in nitrate concentrations was substantially lower than variability in discharge (median CVc/CVq = 0.19, IQR: 0.14 – 0.32). However, QMin-synced catchments showed significantly higher CVc/CVq ratios compared to the other two synchrony types (Fig. S9), suggesting stronger hydrological modulation of nitrate variability in these catchments. Descriptive statistics for discharge, nitrate concentration, and other related variables across all catchments are summarised in Table S1.

Although our classification groups catchments based on their dominant synchrony type, catchments also exhibited varying degrees of interannual variation in their synchrony status (Fig. 2). Among QMax-synced sites, the proportion of QMax-synced sites are the proportion of QMax-synced sites, the proportion of QMax-synced sites are the quarter are the proportion of QMax-synced sites are the quarter are


aligned years ranged from 58% to 90%; similar ranges were found for QMin-synced sites. Asynced catchments showed greater fluctuation between synchrony types.

Figure 3: Annual Regimes of Nitrate Concentrations in (a) QMax-Synced, (b) QMin-Synced Sites (c)Asynced, and (d) Discharge (Q) of all sites. The boxes represent the interquartile range (IQR; 25th-75th percentiles), and whiskers extend to the furthest values within 1.5 x IQR from the box. The central line in each box indicates the median and the smooth connecting curve was generated using B-spline interpolation of the medians.

## 3.2 C-Q relationship and Variablity in Synchrony Patterns

Catchments within each synchrony classification exhibited contrasting nitrate export behaviours, as reflected in the relationship between discharge and peak nitrate concentrations. Here, the interannual slope refers to the regression slope between annual peak nitrate concentrations and their corresponding discharge (in log units) for each catchment (Fig. 4a–c). QMax-synced catchments showed shallower and more variable interannual slopes that were typically weakly positive or near-zero. Almost 42.1% of catchments had a positive and 10.5% exhibited a negative slope.  $\beta_2$  values (i.e. C-Q slopes


estimated from the WRTDS model) of QMax-synced catchments fell mostly in the chemostasis to mobilisation range (Mean  $\beta_2$ : 0.07 ± 0.10).

In contrast, QMin-synced catchments showed the strongest coupling to flow control with consistently steep negative slopes between discharge and peak nitrate, and correspondingly negative  $\beta_2$  values (Mean  $\beta_2$ : -0.42 ± 0.19). Almost 58.8% of catchments had a negative slope between peak nitrate concentrations and corresponding log(discharge). Asynced catchments also exhibited predominantly negative Q–C slopes (Mean: -0.12 ± 0.12), but with greater scatter and a broader mix of  $\beta_2$  values spanning dilution and chemostasis regimes. Nevertheless, many catchments exhibited negative  $\beta_2$  values, implying that their nitrate peaks are often shaped by dilution-like behaviour despite lacking a consistent seasonal alignment with flow.

Despite these differences in export dynamics, peak nitrate concentrations did not differ significantly among synchrony types

Despite these differences in export dynamics, peak nitrate concentrations did not differ significantly among synchrony types (Kruskal-Wallis's test: H = 3.62, df = 2, p = 0.16).

Secondly, we analysed the stability of the timing of nitrate and discharge peaks from year to year. For each synchrony type, we analysed the annual change in the month of peak nitrate concentration and compared it to the change in the month of maximum or minimum discharge (Figs. 4 d&e). In QMax-synced catchments (Fig. 4d), both nitrate and discharge peak timing intervals were mostly stable around the 1:1 line. This temporal coherence is consistent with more reliable nitrate delivery governed by winter flow mobilisation. In QMin-synced catchments (Fig.4e), the timing of minimum discharge varied more substantially between years, yet nitrate peaks generally tracked these changes closely.

Figure 4: Patterns of discharge and nitrate concentrations during peak nitrate months across synchrony types. (a-c) Regression lines are only shown for catchments where the annual regression has  $R^2 > 0.3$ . (d-e) Bubble plots showing the relationship between the interval of peak nitrate months and the interval of maximum or minimum discharge months.

## 245 3.3 Catchment Characteristics controlling Synchrony patterns

Figure 5: Controls of Synchrony Patterns Following the Order of Importance. Boxplots show the distribution of key catchment descriptors ordered by importance from a RF classification model. P values (Wilcoxon rank-sum test) compare QMax- and QMin-Synced Catchment. See Table 1 for descriptions of all variables

To understand which catchment attributes best explain the dominant synchrony behaviour identified in Section 3.2, we used a Random Forest classification model to relate land use, hydrology, and landscape properties to synchrony type (QMax-Synced, QMin-Synced). The random forest classification model achieved a classification error of 12.2%, corresponding to






an overall accuracy of 87.8%. This indicates that the model was able to distinguish between the synchronous patterns based on the selected predictors. Post permutation method (i.e after removal of non-significant variables), only a small increase in classification error was apparent 12.5% (+0.3%).

The most influential catchment descriptors identified by the RF model align with known hydrological and land use controls on nitrate dynamics (Fig. 5). Urban Land area is ranked as the most important factor for the Synchrony patterns classification, followed by the Standard Percentage Runoff (SPR), Arable Land, Baseflow Index (BFI) and the percentage of High Permeable Surface. QMax-Synced catchments had the smallest percentage of urban areas while QMin-Synced catchments had the highest (p < 0.01). In contrast, the percentage of arable land was significantly higher in QMax-Synced catchments compared to QMin-Synced ones. Regarding the hydrological characteristics, linear regression analysis indicated a strong inverse relationship between SPR and BFI, with an R² value of 0.71. The strong linear relationship between SPR and BFI indicated that they essentially represent similar catchment hydrological characteristics. The Lowest SPR and Highest BFI are observed in QMax-Synced catchments. The predictor of High Permeable Surface, reflecting the infiltration capacity and groundwater recharge potential of the catchment, is lowest in QMax-Synced Catchments. Although catchment area did not reach statistical significance (p = 0.07), QMax-synced catchments tended to have smaller catchment areas than QMin-synced catchments.

# 270 3.4 The Drivers of Synchrony Variability

While Section 3.3 identified catchment characteristics that explain dominant synchrony types across space, here we examine what controls interannual variability in synchrony within catchments. For QMax-Synced catchments wetter winters (higher SPI1) were apparent during QMax-Synced years than Asynced years (mean SPI1 difference,  $\Delta = 0.27 \pm 0.32$ , p < 0.01, Table 2). SPI did not impact QMin-Synced catchment behaviour (p > 0.05). In catchments classified as Asynced overall, the years identified as QMin-Synced were significantly wetter, both in terms of winter SPI1 and annual SPI12, than years identified as either QMax-Synced or Asynced (p 


Table 2. Mean differences in Winter SPI1 ( $\Delta \pm SD$ ) and SPI12 ( $\Delta \pm SD$ ) among hydrological year types, based on paired Wilcoxon signed-rank tests within each site.  $\Delta$  represents the mean difference in SPI between synchronous and asynced years at each site.

| Catchment<br>Synchrony<br>Patterns | Difference in (a)synchronous years | n  | Winter SPI1<br>(Δ ± SD)      | p     | SPI12 $(\Delta \pm SD)$ | p     |
|------------------------------------|------------------------------------|----|------------------------------|-------|-------------------------|-------|
| QMax-Synced                        | QMax-Synced vs<br>Asynced          | 16 | $\boldsymbol{0.27 \pm 0.32}$ | <0.01 | $-0.00 \pm 0.53$        | >0.05 |
| QMin-Synced                        | QMin-Synced vs<br>Asynced          | 14 | $-0.10 \pm 0.32$             | >0.05 | $-0.07 \pm 0.41$        | >0.05 |
| Asynced                            | QMax-Synced vs<br>QMin-Synced      | 18 | $-0.45 \pm 0.42$             | <0.01 | $-0.64 \pm 0.62$        | <0.01 |
| Asynced                            | QMax-Synced vs<br>Asynced          | 18 | $-0.09 \pm 0.33$             | >0.05 | $-0.23 \pm 0.56$        | >0.05 |
| Asynced                            | QMin-Synced vs<br>Asynced          | 18 | $0.36 \pm 0.36$              | 

Figure 6: Ternary plots of percentage of synchronous years and key drivers for each catchment, coloured by (a) Arable land percentage, (b) DPS, (c) Urban land percentage, and (d) Population density Density of WWTPs. Coloured: The pink polygon highlights the subset of catchments for which the percentage of the synchronous years was significantly correlated with the corresponding driver (Spearman  $\rho$ , p 







synchrony type in turn, tracing how characteristic source-pathway-connectivity configurations lead to QMax, QMin, or Asynced behaviour.

### 4.1.1 Agriculture-dominated QMax-Synced Catchments

Catchments in which peak flow and peak nitrate concentrations occurred simultaneously (QMax-Synched catchments) exhibited a spatially consistent, agriculture-driven coupling reflecting hydrological mobilisation of ample nitrogen stocks from arable land during periods of higher winter flow and connectivity. These catchments were predominantly distributed in southern and southwestern England accounting for 28.8% of all catchments.

A high percentage of agriculture, specifically arable land cover, was one of the most important factors identified in QMax-Synced catchment. Our analysis also revealed that QMax-Synced catchments were typically smaller headwater catchments with intensive arable land cover. The agricultural area provided sufficient and diffuse nitrogen sources readily mobilised by high flow and younger water, as periods of higher flow enhance catchment connectivity by activating shallow subsurface and surface flow paths that link agricultural areas to the river channel (Yang et al., 2018). This could also be supported by the predominance of chemostasis to mobilisation C-Q relationship in these catchments, which were commonly observed in the previous studies (Zhang, 2018; Moatar et al., 2017) and suggests a uniform distribution of nitrate with depth (Dupas et al., 2016). As highlighted by Worrall et al. (2014), annual maximum nitrate concentrations were sensitive to shifts in nutrient sources such as land use change, fertiliser application. The generally positive or weakly correlated regression slopes observed between annual peak nitrate concentrations and corresponding peak discharges support the ample or non-limiting nitrate pools played an important role in the dominating QMax-synchrony.

The hydrological connectivity also contributed to the synchrony in winter. These agriculturally dominated catchments contained less extensive high-permeability superficial deposits (mainly sands and gravels) and exhibited relatively high Baseflow Index (BFI) and low Standard Percentage Runoff (SPR). This pattern likely reflects the widespread use of artificial subsurface drainage in agricultural areas. After World War II, the UK initiated a major program of land drainage on poorly drained land, which resulted in the delivery of leached nitrate to groundwater and rapidly transported nitrate to streams during rainfall (Green, 1979; Burt et al., 2011). Piped systems were used to drain around 6.4 million hectares of agricultural land in England and Wales (Hill et al., 2018). Such drainage networks could enhance hydrological connectivity and shorten the discharge pathways, promoting the delivery of diffuse nitrogen to streams (Hirt et al., 2005). Tile drains have also been proved to increase the peak flows (Wesström et al., 2001). Thus, the combination of increased hydrological connectivity and sufficient nitrogen supply favoured the occurrence of QMax-Synced catchments. This perspective reframes diffuse agricultural pollution as a timing problem as much as a loading problem. Where increased hydrological connectivity locks nitrate release into the wet (winter) season, management interventions aimed solely at reducing inputs may have limited effect on seasonal synchrony unless they also alter flow–path activation or storage dynamics.






# 4.1.2 Urban-dominated QMin-Synced Catchments

QMin-Synced catchments (24.4 % of the total) exhibited an inverse seasonal regime between nitrate concentrations and discharge and were mainly located in the north-western and southern UK in urban areas. A QMin-Synced pattern has also been reported in some catchments in western France (Guillemot et al., 2021) and the Great Lakes region (Van Meter et al., 2019). This phenomenon can result from strong dilution of stable or legacy nitrate sources (such as groundwater or urban point sources) during high flows (Minaudo et al., 2015), or from the spatial separation between flow-generating areas and nitrogen sources in the catchment (Abbott et al., 2018).

In our study, random forest analysis identified urban area as the strongest explanatory variable in these catchments. Strong dilution patterns were observed and most catchments showed negative regression slopes between peak nitrate concentrations and discharge, suggesting stable urban point sources and lack of large or spatially distributed N stores. Meanwhile, higher SPR and lower BFI, opposite to QMax-Synced catchments, suggest that surface runoff dominated over groundwater contributions, resulting in rapid transport and efficient dilution of nitrate during high-flow periods. Nitrate delivery was tightly coupled to the timing of low flows, i.e. rather than occurring during a fixed month, peak concentrations consistently follow the annual flow minimum.

A legacy-nitrogen explanation for QMin-synchrony, whereby slower drainage of stored soil or groundwater nitrate could elevate concentrations during low-flow periods, would be consistent with the large literature on this topic (Johnson and Stets, 2020). However, our results do not support this mechanism. The strong link to urbanisation with steeply negative C–Q slopes, indicates instead the dominance of stable point inputs rather than gradual legacy release for QMin-synched catchments. Moreover, if both diffuse and point sources were active, we would likely expect dual peaks, one during winter flushing (as with QMax-Synched catchments) and another at low flow, yet only a single low-flow maximum is observed. This pattern further implies that urban infrastructure associated inputs have largely displaced the diffuse, winter-mobilisation behaviour typical of QMax-Synched catchments, creating an engineered inversion where nitrate concentrations peak only under low-flow conditions and are otherwise easily diluted (Kaushal and Belt, 2012; Kaushal et al., 2011).

## 370 4.2 Synchrony Variability and Drivers

Our peak-based analysis showed that, although both nitrate concentration and discharge follow a consistent seasonal cycle on average, the timing of their annual peaks varies substantially among years and catchments. Earlier studies have shown that riverine nitrate concentrations generally track discharge seasonality (Ebeling et al., 2021; Van Meter et al., 2019). Our results extend this understanding by showing that synchrony itself fluctuates through time and space. The strength and timing of nitrate and flow coupling vary with hydro-climatic conditions and catchment characteristics. This variability reveals how climate sets the potential for synchrony, while local land use and hydrological structure determine whether that potential is realised.







# 4.2.1 Winter precipitation and Drainage Path Slope Regulate QMax-Synced Variability

Our results indicated that in QMax-synced catchments, synchrony variability was primarily governed by winter precipitation, nitrogen source availability, and catchment topography. The proportion of QMax years was positively correlated with the fraction of arable land, implying that catchments with larger diffuse N inputs were more likely to exhibit synchronous winter peaks, as nitrate stores in agricultural catchments could be easily mobilised during winter high flow (Jordan et al., 1997; Musolff et al., 2015). Moreover, QMax-Synced catchments experienced significantly wetter winters during synchronous years, as indicated by elevated winter SPI1 values. These wetter winters not only elevated peak discharges in winter but likely also saturating soils and enhancing runoff, thereby increasing the hydrologic connectivity within the catchments. This increased wetness results in shorter hydrological travel times and consequently the co-occurrence of discharge and nitrate peaks (Winter et al., 2022; Blaen et al., 2017). Importantly, the MaxQ percentiles did not differ significantly between QMax and non-QMax years, suggesting that synchrony was not simply driven by more extreme high flows but rather by enhanced connectivity that facilitated nitrogen mobilisation.

While the availability of diffuse nitrogen sources, together with wet winters, sets the stage for synchrony, the efficiency of nitrate flushing was governed by topographic controls. Catchments with lower Drainage Path Slopes (DPS), reflecting flatter topographic gradients and longer flow paths, tended to show weaker and less consistent synchrony between nitrate and discharge peaks. In such settings, slower runoff and longer residence times increase opportunities for nitrate retention, reducing the likelihood that discharge and nitrate peaks coincide. Similar findings have been reported for three German rivers, where flatter topography was associated with reduced N export efficiency (Ehrhardt et al., 2019). In contrast, catchments with steeper drainage paths and rapid runoff generation can facilitate efficient flushing of nitrate from shallow soils during high flow events (Schiff et al., 2002; Harms and Jones, 2012). Overall, our findings extend this previous understanding by showing that, at the seasonal scale, synchrony variability depends less on the magnitude of high flows than on the efficiency with which catchment wetness and structure translate those flows into connectivity. This highlights that nitrate—flow coupling strengthens when climatic wetness aligns with source availability and topographic facilitation of transport, rather than simply when seasonal flow is higher.

# 4.2.2 Low Flow Dilution and Anthropogenic Pressure Drives QMin-Synced Variability

In contrast to QMax-Synced catchments, the synchrony variability in the QMin-Synced were largely shaped by the intensity of anthropogenic loading and the configuration of urban water infrastructure and the temporal variability of discharge in low-flow period. Our analysis showed that more urbanised and densely populated catchments were more likely to exhibit nitrate concentration peaks coinciding with periods of minimum flow. In urban dominated catchments, greater extent of impervious surfaces and engineered drainage systems likely disrupts the natural connection and limit the mobilisation of diffuse sources, especially during wet periods (Duncan et al., 2017). Consequently, during low-flow conditions, persistent point sources like wastewater effluent and sewer leakage can dominate riverine nitrate sources, leading to a stronger

sensitivity of nitrate concentration to dilution effects (Zhao et al., 2023). Meanwhile, The CVc/CVq ratio (generally 







In most Asynced catchments, all three synchrony states were observed across years, highlighting their mixed and shifting controls. This instability likely reflects the absence of persistent hydrological forcing. As we discussed above, QMax-synchrony arose primarily from enhanced hydrologic connectivity during wet winters rather than from the extremity of peak flows, whereas QMin-synchrony depended on sufficient low-flow conditions to emerge. Asynced catchments, likely lacking either consistent high-flow connectivity or prolonged low-flow extremes, are therefore highly prone to switching between synchrony states. Unlike consistently QMin-synced catchments, where continuous anthropogenic loading and extreme low flows dominated and SPI showed little influence, Asynced catchments were more sensitive to climatic anomalies. Wetter winters (higher SPI1) and wetter years (high SPI12) occasionally promoted QMin-like behaviour, though not through urban effluents but likely because increased recharge temporarily reconnected shallow groundwater or legacy nitrate stores to the stream network. It is interesting that these transient connections can mimic the timing of QMin behaviour, nitrate peaks during low-flow periods, but likely arise from hydroclimatic modulation of connectivity rather than from persistent point-source dominance, as they represent the only context in which our analysis indicates a likely contribution from legacy or groundwater-derived nitrate.

Beyond hydroclimatic influences, nitrate export regimes in Asynced catchments were further shaped by catchment size and anthropogenic pressures. Larger catchments tended to dilute diffuse inputs, favouring QMin- like behaviour, while denser wastewater infrastructure reinforced nitrate peaks during low flow periods. Year-to-year differences were also likely influenced by broader environmental variability, such as air temperature, extreme rainfall, or antecedent soil moisture, that can alter uptake and denitrification efficiency and thus accentuate asynchrony (Van Meter et al., 2019). Importantly, because none of these controls acts consistently across years, correlations with individual drivers remain modest. These results indicate that Asynced catchments are not characterised by greater climatic or hydrological variability per se, but by greater sensitivity to it. With weaker structural constraints than the spatially organised QMax and QMin regimes, their synchrony state shifts readily in response to interannual changes in wetness, storage, or loading. This heightened sensitivity may arise because Asynced catchments lack a single dominant source-pathway, as diffuse, groundwater, and urban inputs likely all contribute, but their relative influence depends on hydrological thresholds that vary from year to year. When those thresholds are reached, even modest climatic anomalies can switch the dominant transport pathway, altering whether export resembles QMax- or QMin-like behaviour. The fact that Asynced catchments constitute the most common catchment type in our dataset underscores the importance of this temporally responsive regime, one that is easily overlooked in analyses focused solely on spatial contrasts in land use or source dominance. Together, these results show that the synchrony framework adds a critical temporal dimension to understanding nitrate-flow coupling, revealing how small climatic or infrastructural perturbations can reorganise export dynamics across seasons and years.

https://doi.org/10.5194/egusphere-2025-5130 Preprint. Discussion started: 24 October 2025

© Author(s) 2025. CC BY 4.0 License.

#### **5** Conclusion




We analysed long-term nitrate-discharge seasonality for 66 English catchments to characterise N-Q synchrony patterns and identify the climatic, hydrological, and anthropogenic factors governing this. Three synchrony regimes emerged, QMax-Synced (28.8%), QMin-Synced (25.8%), and Asynced (46.8%) catchments.

QMax-Synced catchments, typically small, agricultural with high base-flow index and low surface permeability, exhibited chemostatic behaviour under high nitrate supply. Synchrony in these catchments was maintained not by the extremity of peak flows but by enhanced hydrologic connectivity during wetter winters, as reflected in high SPI1 values, and especially in catchments with steeper slopes that further promote efficient flushing.

In contrast, QMin-Synced catchments were characterised by higher urban land cover and urban-related point sources, with nitrate peaks predominantly occurring during low-flow periods. Interannual variability in QMin-synchrony was modulated by the interaction between anthropogenic loading, antecedent wetness, and the severity of the low flow extremes. Asynced catchments, the most widespread regime, exhibited frequent interannual switching between synchrony types. This transitional behaviour reflects sensitivity to hydroclimatic anomalies and the interplay between diffuse and point sources, 485 with wetter years or increased effluent inputs favoured QMin-like synchrony, whereas stronger hydrological flushing promoted OMax-like responses.

Overall, our findings demonstrate that peak nitrate-discharge synchrony in catchments is not static but is dynamically regulated by climatic variability, anthropogenic activity and by how these pressures are mediated and expressed through catchment properties. By framing nitrate export in terms of synchrony rather than mean concentration or load, this approach reveals the temporal dimension of catchment response, where the timing and efficiency of connectivity, not just source strength, determine when and how nitrate reaches streams.

## Data availability

Water quality available online Open Water Quality Archive Datasets (WIMS) data were at https://environment.data.gov.uk/water-quality/view/download. Daily discharge records were available from the National River Flow Archive https://nrfa.ceh.ac.uk/. The catchment characteristics were available from https://nrfa.ceh.ac.uk/fehcatchment-descriptors. The Standardized Precipitation Index (SPI) was available from https://ukwrp.ceh.ac.uk/.

#### **Author contributions**

LY, JL, KK and JK conceptualised the research project. LY conducted the formal analysis and prepared the manuscript with contributions from all co-authors.

# 500 Competing interests

At least one of the (co-)authors is a member of the editorial board of Hydrology and Earth System Sciences. The authors have no other competing interests to declare.

# **Acknowledgements:**

LY has been supported by NERC (UK's Natural Environment Research Council) CENTA2 (Central England NERC Training Alliance 2) grant NE/S007350/1. We acknowledge the UK Environment Agency, the National River Flow Archive, and the UK Water Resources Portal for providing essential datasets used in this study. Part of computations described in this paper was performed using the University of Birmingham's BlueBEAR HPC services.

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
