# Peer review of "When does nitrate peak in rivers and why? Catchment traits and climate drive synchrony with discharge"

_EGUsphere, 2025_

## Author Comment (AC1)

**General and specific comments**

I read with great interest the manuscript by Yang et al. which examines the synchrony between the annual peak in nitrate concentration and the maximum or minimum discharge, and the controls of synchrony. The study contributes to a better understanding of the processes and catchment properties that impact nitrate concentration seasonality. However, I think that some clarifications would be needed on several aspects to get the manuscript ready for publication. I summarise my main comments below:

Many thanks for your thoughtful and encouraging comments on our manuscript. We truly appreciate your recognition of this study's contribution to the understanding of the synchrony processes.

**1. The analysis of the synchrony variability between years was not very clear to me.**

Firstly, it would be good to clarify how Qmax-Synced and Qmin-Synced years are distributed in time in the three catchment categories (Qmax-Synced, Qmin-Synced, and Asynced). In other terms are the Qmax-Synced (Qmin-Synced) years grouped together in time and therefore can we separate a time period with a Qmax-Synced (Qmin-Synced) behaviour? Or are there sparsely distributed in time? This would help to understand whether changes in synchrony are due to possible trends or year-to-year variability in drivers, in particular for Asynced catchments.

We agree and thank you for your kind questioning. Regarding the temporal distribution of QMax- and QMin-Synced years, our analysis across all 66 sites shows that synchronous years are quite irregularly scattered with no indication of long-term trends as shown below (Fig. S4). To clarify this, we will attach the figure in the supplements and add the following sentence in the manuscript at the end of the *3.1 Spatial patterns across and temporal variability within catchments*: "Asynced catchments showed greater fluctuation between synchrony types. This interannual variability appears random in time (Fig. S4), with no indication of a long-term trend."

[Figure]

[Figure]

Figure S4 Heatmap of annual synchrony types (2001–2019) across 66 catchments

Secondly, I have been a bit confused regarding the analyses presented in Table S11 and the pink polygon of Fig. 6 and I think that their meaning should be clarified in the manuscript. From my understanding, these results refer to differences between catchments and are not explaining the year to year variability of synchrony given the catchment. To me, only the analyses based on precipitation and discharge (L271-286) examine what drives the temporal variability in synchrony.

We agree and thank the reviewer for this helpful comment. The analyses presented in Table S11 and the pink polygon of Fig. 6 were included to help explain variability in synchrony composition. Here, our intention was to address both spatial synchrony variability (why certain catchments tend to be QMax-Synced, QMin-Synced, or Asynced) and temporal variability (why specific years fall into these categories within a catchment). We agree this distinction should be made more explicit.

To clarify this separation in the manuscript, we will add the following sentence at the beginning of this section: "In addition to the interannual variability, we also considered spatial variability in synchrony composition captured as long-term differences in the relative occurrence of QMax-, QMin- and Asynced years across catchments. These spatial correlations identify why some catchments tend to exhibit more QMax-Synced, QMin-Synced, or Asynced years over the 2001–2019 period."

**2. A few methodological points would require clarification regarding the catchment properties used and the time scale of the analysis.** Was synchrony determined based on concentration and discharge data at daily time scale or aggregated to monthly time scale?

We agree and many thanks for this useful comment. Our synchrony was determined at the monthly scale. Daily nitrate concentrations were first reconstructed using WRTDS and aggregated to monthly mean concentrations, consistent with the aggregation of daily discharge to monthly mean discharge. The classification of QMax-Synced, QMin-Synced and Asynced years is therefore based on the timing of monthly concentration and monthly discharge peaks. We will clarify this more explicitly in the Methods section 2.2.

We will add the following sentence after the L115: "For the synchrony analysis, both reconstructed concentrations and observed discharge were subsequently aggregated to monthly mean values, and all synchrony classifications were determined at the monthly scale based on the timing of monthly maxima and minima."

**3. I think that the background should be expanded in the introduction (see details in the following).**
I provide in the following detailed comments below that I hope will be of help to the authors to revise the manuscript.

- P2 L44: this should be nuanced and better discussed, since C-Q relationships can be used for analyses at different temporal scale, as highlighted for instance in Musolff et al. (2021).

This is an important point and we agree that that C–Q relationships can be applied across multiple temporal scales, as demonstrated in Musolff et al. (2021). Our intention in the original sentence was not to undervalue C–Q relationships, but to highlight that, when applied in their conventional form, they primarily characterise magnitude-based concentration–flow behaviour and may not explicitly capture the timing alignment, which is central to our synchrony framework.

To better reflect this and improve nuance, we will revise the sentence: "While concentration–discharge (C–Q) relationships offer valuable insights into source–mobilisation dynamics and hydrological controls at different temporal scales (Musolff et al., 2015; Bieroza et al., 2018; Knapp and Musolff, 2024), they primarily describe magnitude-based responses. When interpreted alone, they may not resolve whether the timing of concentration peaks aligns with hydrological extremes, thereby obscuring temporal shifts in nitrate delivery pathways."

- P2 L51-52: what are these studies, what do we learn from them, and why are they not sufficient?

Many thanks for this helpful comment. Van Meter et al. (2020) and Abbott et al. (2018) were cited as examples of how seasonal alignment or misalignment between hydrological forcing and biogeochemical responses can reveal important differences in catchment functioning. However, both studies focus on average seasonal behaviour derived from multi-year mean cycles. They did not quantify whether the timing of peak concentrations aligns with peak or minimum discharge from one year to the next, nor did they examine how this alignment changes under interannual climatic variability. This interannual timing dimension is the gap our synchrony metric is designed to fill.

To make this clearer, we will add the sentence after L52 as follows: "These studies demonstrate that the degree of seasonal alignment reflects how source availability, connectivity, and hydrological forcing interact to shape nutrient export. However, because they rely on multi-year average seasonal regimes, they may not further explore whether the timing of nitrate and discharge peaks remains aligned from year to year and how this alignment responds to climatic variability."

- P2 L55 "data gaps": I do not understand this, as the annual peak may not be identifiable because of data gaps. This would need to be further explained.

We agree and thank the reviewer for raising this point. Our intention here was not to imply that synchrony is unaffected by missing data, but rather that it is appropriate for low-frequency and irregular water-quality datasets. Synchrony requires only the identification of the month of peak concentration or discharge. It could be reliably estimated provided that monthly coverage is reasonably complete. This is also why we required a threshold of 80% monthly coverage in our data screening.

To avoid misunderstanding, we will revise the sentence to clarify this point: "By focusing on the timing of annual peaks, the synchrony metric does not depend on short-term fluctuations or event-scale variability, making it suitable for low-frequency and irregularly sampled datasets. It only requires sufficient monthly coverage to identify peak timing."

- P2 L62: I think it would be good to discuss previous literature that analysed the drivers of concentration (beyond discharge) to clarify and highlight the contributions of this study.

We agree and thank the reviewer for this helpful suggestion. We agree that, in addition to discharge, a range of biogeochemical, landscape, and climatic factors are known to influence nitrate concentration.

A new paragraph will be added in the section 1 of the revised version to highlight our contribution more explicitly: "Beyond flow control, riverine nitrate concentrations are also influenced by multiple factors. Nitrogen inputs from fertiliser, manures and wastewater largely determine baseline nitrate levels, while landscape and hydrogeological properties, such as soil permeability, groundwater pathways and denitrification capacity, govern N retention and subsurface delivery (Ehrhardt et al., 2021; Jiajia et al., 2021). Meteorological controls like moisture conditions, drought–rewetting cycles, and recharge pulses further modulate mineralisation, leaching, and dilution, generating substantial temporal variability even under similar land-use pressures (Mcaleer et al., 2022). These controls describe why nitrate levels rise or fall, but not when peak nitrate delivery occurs, a key dimension for management. Our synchrony analysis explicitly targets this temporal gap."

- Table 1:

Why was SPI1 adopted for winter months only and not for summer months as well?

We appreciate the reviewer's question regarding the use of winter SPI1. We focused on winter SPI1 because hydrological connectivity in UK catchments is strongly seasonal. Summer rainfall rarely produces catchment-scale connectivity due to high evapotranspiration and large soil-moisture deficits; most summer precipitation is absorbed into dry soils rather than generating runoff or activating shallow subsurface pathways (Kilsby et al., 2019; Barker et al., 2016). In contrast, winter rainfall is the dominant driver of connectivity in temperate UK catchments because soils are near saturation and both surface and subsurface flow paths are fully activated

(Muchan et al., 2015). For these reasons, we used winter SPI1 for analysing QMax-synchrony and the longer-term SPI12 to characterise persistent moisture deficits relevant to QMin-synchrony.

We will clarify this rationale in the revised manuscript L105: "Winter SPI1 was used because winter precipitation is the main driver of hydrological connectivity in UK catchments, when soils approach saturation and both surface and subsurface pathways become activated (Muchan et al., 2015). In contrast, summer flows are controlled largely by evapotranspiration and accumulated soil-moisture deficits (Kilsby et al., 2019), making SPI12 a more appropriate indicator of antecedent conditions relevant to low-flow behaviour."

Are the different catchment properties static or time-varying? My guess given the rest of the manuscript is that all properties are static but SPI. This point links to the variability analysis presented in Sect. 3.4: would it be possible to relate the synchrony variability between years to the variability in catchment properties beyond SPI and discharge?

We thank the reviewer for this thoughtful question. We agree that linking synchrony variability to time-varying catchment properties would be an important extension. In our study, descriptors except SPI (e.g., BFI, PROPWET, DPS, SPR, WWTP density) were treated as static structural properties to represent the structural characteristics of the catchments. As highlighted by Ehret et al. (2014), catchment structure changes slowly over decadal scales, whereas hydro-climatic forcing (precipitation, wetness anomalies) operates at much shorter time scales and is the dominant source of interannual variability. In addition, robust and consistently resolved annual time series for many structural properties are not available across the full monitoring period and all catchments, which limits their reliable inclusion. We therefore focus on hydro-climatic variability as the primary control on interannual synchrony in this study. Investigating how gradual changes in catchment structure interact with climate forcing to shape synchrony dynamics would be an important direction for future work.

Please add a column for the data source.

Many thanks for this useful suggestion. We will add a new column for the sources in Table 1.

It would be helpful to provide further details (equation) on the hydrological variables (this could be in appendix or supplements).

We agree and thank the reviewer for this helpful suggestion. In the revised manuscript, we will add a new section in the Supplementary Material that provides clearer definitions and some mathematical formulations as follows:

**S1. Hydrological Variable Definitions and Equations**
The mathematical definitions and calculation procedures for all hydrological variables used in the analysis are summarised below. Equations are based on the Flood Estimation Handbook (FEH) Volume 5 (Centre for Ecology & Hydrology, 1999) and values were obtained directly from the National River Flow Archive (NRFA).

S1.1 Baseflow Index (BFI)

BFI quantifies the proportion of streamflow contributed by groundwater and delayed pathways. In this study, BFI values were obtained directly from the National River Flow Archive (NRFA), where they are computed from observed daily discharge using the UKIH smoothed minima baseflow separation method (Gustard et al., 1992).

$$BFI = \frac{Q_{baseflow}}{Q_{total}}$$

where

$Q_{baseflow}$ is the flow component extracted following the UKIH smoothed minima algorithm,

$Q_{total}$ is total annual discharge.

S1.2 Proportion of Time Catchment Soils are Wet (PROPWET)

PROPWET measures how frequently catchment soils are above a wetness threshold.

$$PROPWET = \frac{1}{T} \sum_{t=1}^{T} I(\theta_t > \theta_{wet})$$

where

$\theta_t$ = modelled soil moisture at time $t$,

$\theta_{wet}$ = FEH-defined wetness threshold corresponding to near-saturated conditions,

$I(\cdot)$ = indicator function.

Values close to 1 indicate persistently wet soils ; lower values indicate predominantly dry soils.

S1.3 Flood Attenuation by Reservoirs and Lakes (FARL) index

FARL quantifies the cumulative attenuation effect of on-line lakes and reservoirs on high flows. Values near 1 indicate negligible attenuation; smaller values indicate strong attenuation.

$$FARL = \prod_{i=1}^{n} (1 - \sqrt{\frac{A_{lake,i}}{A_{subcatch,i}}})^{\frac{A_{subcatch,i}}{A_{catch}}}$$

where

$n$ = number of on-line lakes or reservoirs in the catchment.

$A_{lake,i}$ = surface area of lake/reservoir $i$

$A_{subcatch,i}$ = upstream drainage area contributing to lake/reservoir $i$.

$A_{catch}$ = total catchment area at the outlet.

S1.4 Standard Percentage Runoff (SPR)

SPR (specifically SPRHOST) expresses the standard percentage of rainfall that appears as direct (quick) runoff under average conditions. It is estimated in FEH from the distribution of HOST (Hydrology of Soil Types) soil classes within the catchment.

S1.5 Drainage Path Slope (DPS, m km$^{-1}$)

DPS (specifically DPSBAR) describes the mean slope along the drainage paths defined by the Integrated Hydrological Digital Terrain Model. For each grid cell, the steepest descent direction to a neighbouring

cell is used to define a local slope; DPS is then obtained as the catchment-wide mean of these drainage-path slopes. Higher DPS values correspond to steeper, more rapidly draining catchments.

- P6 L148: the p-value of which test is this?

Thank you for pointing out this ambiguity. The p-value refers to the empirical permutation-based significance of variable importance following Altmann et al. (2010), where the null distribution of importance scores is generated by repeatedly permuting the response variable. We will revise the text to clarify that this is not a parametric statistical test, but an empirical p-value derived from the permutation procedure.

We will revise the sentence as follows: "Following cross-validation, a final RF model was trained on the full dataset, and permutation-based variable importance values were extracted from the underlying ranger model, following Altmann et al. (2010). Empirical permutation p-values (<0.05) were obtained from 1000 random permutations of the response variable to identify descriptors whose importance exceeded the null distribution."

- P7 L165: I think that the term "synchrony variability" should be better defined.

We agree and thank the reviewer for this helpful suggestion. We now clarify this "synchrony variability" further refers to a spatial metric quantifying how catchments differ in their long-term tendency toward QMax-, QMin-, or Asynced synchrony over the 20-year period, reflecting spatial differences in the stability of synchrony behaviour across catchments.

The revised sentence in the Methods will read: "Last, we used Spearman rank correlations and non-parametric Wilcoxon rank-sum tests to assess the relationship between spatial synchrony variability, defined as the degree to which catchments differ in their long-term tendency towards different synchrony, and potential influential factors, calculated and visualised using the numpy, pandas, seaborn, matplotlib and scipy packages in Python."

- P7 L170-171: I can see that the performance is rather low for some catchments (R2 as low as 0.11), which indicates that the processed data should be used with care. However, in the end, the actual values of the concentration is not the main focus but its synchrony. I am therefore wondering whether another performance metric, that would focus on the temporal pattern, could be relevant to complement the performance analysis (such as Spearman or Pearson correlation).

We agree and thank you for this valuable suggestion. We therefore evaluated model performance using monthly Spearman's rank correlations between observed concentrations and WRTDS-reconstructed values across all catchments. The results show that the temporal structure is well preserved in the reconstructed series (median Spearman's $\rho$ = 0.72, IQR = 0.134).

We'll add a new sentence in the beginning of Result: "To assess the temporal consistency, spearman correlations between observed and reconstructed concentrations were calculated showing a median $\rho$ of 0.72 (IQR = 0.134). No sites were excluded based on model performance." And we will revise the sentences in the methods: "For the synchrony analysis, both reconstructed concentrations and observed discharge were subsequently aggregated to monthly mean values, and all synchrony classifications were determined at the monthly scale based on the timing of monthly maxima and minima. To assess the adequacy of WRTDS-

estimated concentrations for identifying monthly peaks, we computed the temporal Spearman correlation between observed and modelled monthly concentrations to assess agreement in temporal patterns relevant to the synchrony analysis."

- P11 L228: how is the change in peak nitrate concentration and discharge calculated? From one year to the next?

Thank you for the comment. Here, the change refers to the year-to-year difference in peak timing between consecutive years. We will adjust the sentence to clarify as follows: "For each synchrony type, we analysed the consecutive-year change in the month of peak nitrate concentration and compared it with the corresponding change in the month of maximum or minimum discharge (Fig. 4 d&e)."

- P11 L229-232: From Figure 4, I see that the changes in the timing of discharge are rather small for both Qmax-synced and Qmin-synced catchments (mostly between +1 and -1). Perhaps the only notable difference is that changes in concentration seem to more systematically follow changes in Q for Qmin-synced catchments?

We agree and thank the reviewer for this insightful observation. We agree that interannual changes in discharge peak timing are small for both synchrony types, with most shifts falling within ±1 month. Figure 4d shows that QMax-synced catchments exhibit tightly clustered and highly stable peak timing, with nitrate peaks predominantly occurring in winter. In contrast, minimum-flow timing varies more widely across years and seasons in QMin-synced catchments (Fig. 4e), and year-to-year shifts in nitrate peak timing more systematically follow the corresponding changes in $\Delta Q$.

We will clarify as follows: "In QMax-synced catchments (Fig. 4d), both nitrate and discharge peak timing intervals were mostly stable around the 1:1 line, with most intervals confined within ±1 month. This temporal coherence is consistent with more reliable nitrate delivery governed by winter flow. In QMin-synced catchments (Fig.4e), the interval between minimum-flow months varied more widely across years and seasons, yet the corresponding intervals between nitrate peaks tracked these changes closely."

- Figure 4.d: there is an error in the x-axis, the middle value should be 0 instead of -4.

Thank you for pointing this out. The x-axis label in Figure 4d will be amended with the central tick now correctly displaying 0.

[Figure]

Figure 4: Patterns of discharge and nitrate concentrations during peak nitrate months across synchrony types. (a-c) Regression lines are only shown for catchments where the annual regression has R² > 0.3. (d-e) Bubble plots showing the relationship between the interval of peak nitrate months and the interval of maximum or minimum discharge months.

- P14 L286 "median 0.086 vs 0.068, p=0.007": the difference appears to be rather small and I think that the results should be more nuanced. A low p-value only means that we can distinguish the two values given the sample size, but it does not mean that the difference is actually large and relevant for the analyses.

We agree and thank the reviewer for this helpful comment. We agree that the numerical difference between the two medians is small and that a low p-value does not imply a large effect size. We will revise the sentence as follow to better reflect this nuance: "In contrast, in catchments where QMin-synchrony was more frequent (30 sites), non-QMin years showed slightly higher percentiles of the minimum-flow month than QMin years (median 0.086 vs 0.068). Although the difference is modest, it was statistically significant (p = 0.007)."

- P15 L293-302: It was not easy to get around these analyses. I suggest to better guide the reader through the different figures. In particular, I understand that the analyses refer to the pink polygon of Fig. 6 (this should be explained) and the extended results of the correlation analysis in Fig. S11. In addition, the caption of Fig. S11 is not very explicit and should be revised. My understanding is that the correlation is calculated between the catchment

properties and their percentage of synchronous years (using one value for each catchment), while the caption suggests that a value for each catchment and each year is used. But maybe I missed something?

Many thanks for pointing out. We agree that the connection between fig.6 and the correlation analysis required clearer explanation. We will clarify that the correlations were calculated at the catchment level, using each catchment's proportion of QMax-, QMin- and Asynced years and then conducted spearman correlations within each synchrony group.

We'll modify the paragraph to clarify our results as follows: "In addition to the interannual variability, we also considered spatial variability in synchrony composition captured as long-term differences in the relative occurrence of QMax-, QMin- and Asynced years across catchments. These catchment-level synchrony metrics were then related to catchment attributes to identify the key controls on the dominant synchrony state (i.e. providing insight into why certain catchments display QMax-, QMin- or mixed synchrony). Within QMax-synced catchments, a higher share of QMax years was associated with greater arable land cover ($\rho$ = 0.58, p < 0.05) and lower Drainage Path Slope (DPS; $\rho$ = −0.50, p < 0.05). In contrast, in QMin-synced catchments, urban land cover ($\rho$ = 0.53, p < 0.05), population density ($\rho$ = 0.58, p < 0.05), as well as higher CVc/CVq and The Proportion of Time Soils Are Wet (PROPWET) (both $\rho$ = 0.58, p < 0.05), were all positively associated with the proportion of QMin-synced years. Asynced catchments behaved as transitional systems with a higher density of WWTPs shifting the year mix towards QMin-Synced years ($\rho$ = 0.36, p < 0.05) and away from QMax-Synced years ($\rho$ = -0.50, p < 0.05). A higher baseflow index (BFI) tended to support QMax-Synced years ($\rho$ = 0.38, p<0.05), while larger catchment area ($\rho$ = 0.39, p < 0.05), and lower SPR ($\rho$ = -0.38, p < 0.05) were linked to a greater prevalence of QMin-Synced years. A full summary of additional correlations is provided in Fig. S7."

And the Figure will be modified as follows: "Figure S7 Heatmap of Spearman correlations (p < 0.05) between catchment descriptors and the long-term proportion of QMax-, QMin- and Asynced years."

- P15 L293: SPR as well, no?

We agree and thank you for the helpful suggestion. In the revised manuscript, we will explicitly include SPR among the significant correlates as follows: "Within QMax-synced catchments, a higher share of QMax years was associated with greater arable land cover ($\rho$ = 0.58, p < 0.05), lower Drainage Path Slope (DPS; $\rho$ = −0.50, p < 0.05) and low standard percentage runoff (SPR, $\rho$ = -0.53, p < 0.05)."

- Figure 6: Which catchments are considered? I can see less than 66 data points.

We thank the reviewer for the observation. Figure 6 includes all 66 catchments used in the analysis. The apparent number of points is lower because several catchments share similar synchrony compositions and therefore overlap in the ternary space. Thus, we'll replace it with the figure below:

[Figure]

Figure 6: Ternary plots of percentage of synchronous years and key drivers for each catchment, coloured by (a) Arable land percentage, (b) DPS, (c) Urban land percentage, and (d) Population density Density of WWTPs. Coloured: The pink polygon highlights the subset of catchments for which the percentage of the synchronous years was significantly correlated with the corresponding descriptor (Spearman ρ, p < 0.05).

- P16 "temporal reorganisation of those same controls": This is not fully clear to me, since, to me, the temporal variability in land use, geology and drainage infrastructure was not really analysed in the manuscript but only their differences between catchments (see my main comment 1) above.

We appreciate the reviewer's clarification. We agree that the previous wording was ambiguous and could be misinterpreted. We will revise the sentences L340 only clarifying the mixed Asynced synchrony in the stage of spatial analysis, as follows: "Asynced catchments, in contrast, lack a single dominant source–pathway configuration, resulting in a mixed synchrony signature."

- Sect. 4.1.2: This section could be more concise. In particular, legacy is discussed at several locations (L351, 361). I think that points on the same idea should be grouped together.

Thank you for the helpful suggestion. We will streamline the first paragraph by removing mechanism-related explanations and retaining only a concise description of the observed QMin-synchrony pattern and the relevant literature reporting similar behaviour. All mechanism discussion is now consolidated in the subsequent paragraphs.

The revised first paragraph will be as follows: "QMin-Synced catchments (24.4 % of the total) exhibited an inverse seasonal regime between nitrate concentrations and discharge and were mainly located in the north-western and southern UK in urban areas. A QMin-Synced pattern

has also been reported in some catchments in western France (Guillemot et al., 2021) and the Great Lakes region (Van Meter et al., 2019)."

- Sect. 4.2: With reference to my main comment 1), I understand that this section discusses in part the variability in space that was already discussed in Sect. 4.1. This creates redundancies.

We agree and thank the reviewer for this comment. We will revise Section 4.2 to clearly separate it from the spatial analysis in Section 4.1. In this section, we will focus on how interannual climate variability regulates synchrony within each catchment, and how certain spatial attributes modulate the sensitivity of catchments to these year-to-year hydro-climatic changes (see below).

Specifically, Section 4.2.1 will focus on how winter precipitation regulates year-to-year variability in QMax-synchrony, with catchment properties (e.g. drainage Path Slope) discussed only in terms of how they modulate this climatic sensitivity. Section 4.2.2 similarly will focus on how low-flow severity drives interannual variability in QMin-synchrony, with spatial attributes such as urbanisation and antecedent wetness framed as controls on catchment responsiveness rather than as primary spatial comparisons.

- P20 "stronger and more dynamic anthropogenic pressures": isn't this in contradiction with p9 L200 ("stronger hydrological modulation of nitrate variability in these catchments")? Do urban nitrate sources (such as wastewater effluents) really have strong dynamics?

We thank the reviewer for highlighting this inconsistency. We agree that wastewater effluent and sewer leakage represent persistent rather than dynamic anthropogenic inputs, and therefore the original wording was a little misleading. In Section 4.2.2 we further showed that CVc/CVq is positively associated with the proportion of QMin-synchrony years across sites. We interpret this relationship as evidence that more urbanised catchments are more sensitive to low-flow extremes.

To avoid the unintended implication, we'll revise the sentence to: "More urbanised catchments exhibited stronger hydrological modulation of nitrate concentrations during severe low-flow conditions, as reflected in the positive association between CVc/CVq and the proportion of QMin-synchrony years."

- P21 L 455 "shaped by catchment size": where is this results shown?

Thank you for pointing this out. The influence of catchment size on synchrony patterns is reported in Section 3.4, where we analysed catchment-level Spearman correlations. Specifically, larger catchment area was positively correlated with the proportion of QMin-synchrony years ($\rho = 0.39$, $p < 0.05$). Asynced catchments represent transitional systems whose year-to-year synchrony outcome shifts between QMax- and QMin-type years. this association implies that catchment size contributes to shaping the mixture of synchrony modes expressed in Asynced systems.

- P21 L457-459 and L463-465: would you have sufficient data to test this in the manuscript? or could you discuss what you would need?

We agree and thank you for this insightful comment. The mechanisms proposed in these sentences such as extreme rainfall, antecedent soil moisture, are based on existing understanding from previous studies but may not be tested directly in this manuscript. Our analysis offers a first national-scale characterisation of nitrate synchrony regimes and their potential drivers, relying on a combination of static catchment descriptors and some long-term hydroclimatic time series data. We need more efforts to integrate multiple time series data sources. But this study is an important first step to explore which controls are most relevant across diverse catchments and provides a coherent framework that incorporates timing into C–Q relationships. Moving forward, we would like to collect more data involving slowly evolving variables like land-use, soil and subsurface characteristics, and also drainage infrastructure, fluctuating climatic and hydrological data such as rainfall intensity, soil moisture dynamics, groundwater levels and even some biogeochemical observations such as whole system metabolism or denitrification rates. Such datasets would enable future work to build more integrated models that resolve how fast climatic fluctuations interact with slowly varying catchment structure to influence interannual synchrony dynamics.

References:

Altmann, A., Toloşi, L., Sander, O., and Lengauer, T.: Permutation importance: a corrected feature importance measure, Bioinformatics, 26, 1340-1347, 10.1093/bioinformatics/btq134, 2010.

Barker, L., Hannaford, J., Muchan, K., Turner, S., and Parry, S.: The winter 2015/2016 floods in the UK: a hydrological appraisal, Weather, 71, 324-333, 10.1002/wea.2822, 2016.

Ehret, U., Gupta, H. V., Sivapalan, M., Weijs, S. V., Schymanski, S. J., Blöschl, G., Gelfan, A. N., Harman, C., Kleidon, A., Bogaard, T. A., Wang, D., Wagener, T., Scherer, U., Zehe, E., Bierkens, M. F. P., Di Baldassarre, G., Parajka, J., van Beek, L. P. H., van Griensven, A., Westhoff, M. C., and Winsemius, H. C.: Advancing catchment hydrology to deal with predictions under change, Hydrol. Earth Syst. Sci., 18, 649-671, 10.5194/hess-18-649-2014, 2014.

Ehrhardt, S., Ebeling, P., Dupas, R., Kumar, R., Fleckenstein, J. H., and Musolff, A.: Nitrate Transport and Retention in Western European Catchments Are Shaped by Hydroclimate and Subsurface Properties, Water Resour. Res., 57, 10.1029/2020wr029469, 2021.

Jiajia, L., Compton, J. E., Hill, R. A., Herlihy, A. T., Sabo, R. D., Brooks, J. R., Weber, M., Pickard, B., Paulsen, S. G., and Stoddard, J. L.: Context is Everything: Interacting Inputs and Landscape Characteristics Control Stream Nitrogen, Environ. Sci. Technol., 55, 7890-7899, 10.1021/acs.est.0c07102, 2021.

Kilsby, C., Fowler, H., Lewis, E., Archer, D., and Ledingham, J.: Contrasting seasonality of storm rainfall and flood runoff in the UK and some implications for rainfall-runoff methods of flood estimation, Hydrol. res., 50, 1309-1323, 10.2166/nh.2019.040, 2019.

McAleer, E., Coxon, C., Mellander, P.-E., Grant, J., and Richards, K.: Patterns and Drivers of Groundwater and Stream Nitrate Concentrations in Intensively Managed Agricultural Catchments, Water, 14, 10.3390/w14091388, 2022.

Muchan, K., Lewis, M., Hannaford, J., and Parry, S.: The winter storms of 2013/2014 in the UK: hydrological responses and impacts, Weather, 70, 55-61, 10.1002/wea.2469, 2015.

Spill, C., Ditzel, L., and Gassmann, M.: In-Stream Nitrogen Dynamics in a Point Source Influenced Headwater Stream During Baseflow Conditions, Water Resour. Res., 60, 10.1029/2023wr036672, 2024.

---

## Author Comment (AC2)

**Review Comments**

This manuscript by Yang et al. explores the association between land use, hydrological, and climate properties on watershed nitrate export. They mainly explore the spatial and temporal patterns of synchrony between annual peak nitrate concentrations and peak flow, with the aim of understanding the typologies of watersheds where peak N aligns with maximum discharge (QMax-Synced), minimum discharge (QMin-Synced), or neither (Asynced). I believe that this work contributes to a better understanding of nitrate export across land-use gradients, and the exploratory nature of the paper provides a foundation for future investigation of the mechanistic drivers of watershed seasonality patterns. However, I believe revisions are needed to make the manuscript publication ready.

We thank the reviewer for the constructive and encouraging comments on our manuscript. We are pleased that you recognise the contribution to improving understanding of nitrate export across land-use and hydro-climatic gradients, and that the synchrony framework provides a useful basis for mechanistic investigation.

**General comments**

- The current analyses using descriptive statistics and RF model are finding correlations and associations between catchment feature and synchrony class. Throughout the manuscript and in the title, "drivers" is used which implies a mechanistic association. I think it would be more accurate to use "associations" or "correlations" in place of "drivers".

  We agree and thank the reviewer for this helpful point. We will revise the title as: "When does nitrate peak in rivers and why? Catchment traits and climate relate to synchrony with discharge".

- Please clarify there was any pre-processing and outlier detection of the water quality data. For example, NW-88004024 with $R^2 = 0.11$ appears to be influenced by a single extreme outlier (~16 mg/L when all other observations are <4 mg/L). Is this observation is reliable? Additionally, was there any gap filling of flow in the cases where flow data was missing? Was there a screening for catchments based on proportion of high flow events that had sampled N data?

  We thank the reviewer for this insightful comment. As correctly pointed out, $R^2$ is highly influenced by outliers. Following suggestions from reviewer 1, we will present Spearman's Rho instead of R2 in the revised manuscript. The results show that the temporal structure is well preserved in the reconstructed series (median Spearman's $\rho$ = 0.72, IQR = 0.134).

  We applied the threshold of 90% available over the 20-year period on screening the flow data mentioned in the Method 2.1. Short missing segments were filled using simple linear interpolation to produce the continuous daily series.

  We'll add a new sentence in the beginning of Result: "To assess the temporal consistency, spearman correlations between observed and reconstructed concentrations were calculated showing a median $\rho$ of 0.72 (IQR = 0.134). No sites were excluded based on model performance." And we will revise the sentences in the methods: "For the synchrony analysis, both reconstructed concentrations and observed discharge were subsequently aggregated to monthly mean values, and all synchrony classifications were determined at the monthly scale based on the timing of monthly maxima and

minima. To assess the adequacy of WRTDS-estimated concentrations for identifying monthly peaks, we computed the temporal Spearman correlation between observed and modelled monthly concentrations to assess agreement in temporal patterns relevant to the synchrony analysis."

We will add the gap filling method about the discharges on L96: "Short missing segments of daily discharges were filled using simple linear interpolation to produce the continuous daily series."

Station NW-88003442 exhibits markedly higher concentration variability before ~2004 compared to after, suggesting a potential shift in catchment or monitoring conditions. WRTDS assumes gradual evolution of concentration-discharge relationships and may not perform well when abrupt changes occur (Hirsch et al., 2010). The authors should investigate whether a known change (e.g., dam construction, wastewater treatment upgrades, monitoring protocol change) occurred around this time, and discuss whether modeling this site as a single continuous record is appropriate or whether the pre- and post-2004 periods should be treated separately. While model performance statistics suggest WRTDS handled this transition adequately, if the catchment underwent a structural change, the extracted $\beta_2$ coefficients and synchrony classification for this site may reflect a blend of two distinct periods rather than a coherent long-term signal?

We agree and many thanks for this thoughtful comment. The site NW-88003442 shows a structural change in nitrate behaviour around 2003–2004. However, based on available public records, we were unable to identify a clearly attributable intervention.

To assess the impact on our synchrony results, we re-computed peak concentration months and synchrony classification using only the post-2004 data. The result of synchrony categories is also consistent with the previous result, which does not alter any of the aggregated patterns or conclusions reported in the manuscript. We therefore retain the full record in the main analysis for consistency across sites and add the sensitivity analysis for this site.

We will include the sentence at the end of Method 2.3: "One catchment (NW-88003442) exhibited a clear structural change in the early 2000s. As a robustness check, we recalculated its synchrony using only post-2004 data; the dominant synchrony category remained unchanged, so the full record was retained for consistency across sites."

Lines 190-194, 225-226: The finding that concentration magnitudes across Qmax and Qmin classes do not differ is surprising given that arable land, and thus I assume nitrogen inputs, is significantly higher in QMax-Synced catchments. This is not a typical pattern I've seen across other catchments. Some discussion of why in these catchments higher agricultural inputs don't translate to higher concentrations would strengthen the interpretation.

We agree and thank the reviewer for this comment. The observation that peak concentrations do not differ significantly across synchrony classes, despite higher arable cover in QMax-Synced catchments, is indeed important and requires further clarification. Across many UK studies, river nitrate concentrations have been shown to decouple from agricultural nitrogen inputs because winter high-flow conditions, groundwater mixing, and subsurface denitrification collectively suppress concentration peaks even in highly arable catchments (Bowes et al., 2014; Hiscock et al., 2023). By contrast, although their overall nitrogen inputs are lower, QMin-Synced catchments may reach nitrate peaks comparable to, or even exceeding, those in

agricultural QMax catchments because summer low-flow conditions sharply reduce dilution and amplify the influence of continuous urban and wastewater inputs (Cooper et al., 2022).

We will add following sentence to the Discussion 4.1 : "Although agricultural catchments receive larger nitrogen inputs, winter high-flow conditions, groundwater mixing and subsurface denitrification can suppress concentration peaks (Bowes et al., 2014; Hiscock et al., 2023), whereas in urban catchments summer low-flow periods amplify continuous wastewater and urban drainage inputs (Cooper et al., 2022); as a result, QMin-Synced catchments may exhibit peak nitrate concentrations comparable to those in QMax-Synced systems"

The interpretation of QMin-Synced catchments as uniformly "urban-dominated" may oversimplify what appears to be a mechanistically heterogeneous group. I understand that you are looking at dominant behavior and broad stroke patterns, however, I think attributing QMin-Synced catchment behavior exclusively to urban dynamics leads to missing important nuance. For instance, while arable land is significantly higher in QMax-Synced catchments on average, Figure 5 shows that approximately 25% of QMin-Synced catchments have >20% arable land cover. Depending on agricultural intensity, proximity to the watershed outlet, etc., this fraction could meaningfully contribute to nitrate dynamics. The possibility that a subset of QMin catchments reflects agricultural legacy contributions, particularly in catchments with lower drainage density, deserves consideration before concluding that QMin synchrony is exclusively urban-driven. Consider stratifying QMin-Synced catchments by WWTP density to distinguish (1) High-WWTP catchments, high urban land use, and high population density where urban point sources likely dominate, and low-WWTP catchments where other mechanisms (legacy groundwater, forested catchment dynamics, or non-WWTP urban sources) may drive QMin behavior.

We thank the reviewer for this thoughtful insight and suggestion to consider the heterogeneity within QMin-Synced catchments. Additional Spearman analyses were conducted focusing specifically on QMin catchments. These analyses show that QMin catchments form a continuous land-use gradient but no discrete subgroups. Median nitrate concentrations may increase with arable cover ($\rho = 0.41$, p=0.11) and urban land ($\rho = 0.42$, p=0.10) and decrease with grassland cover (–0.50, p ≤ 0.05), indicating that background concentration levels are affected primarily by land use composition. The reason why we have not used the WWTPs density is because it does not directly correlate with urban land cover or population density. High densities of treatment work often occur in rural areas, whereas large cities are often served by a small number of large capacity treatment works.

The manuscript states that higher $CV_C/CV_Q$ in QMin catchments signals "stronger and more dynamic anthropogenic pressures." and "increased urbanisation and population density are likely the main drivers of QMin-Synchrony, reflecting the dominance of continuous anthropogenic nitrate inputs." The first sentence is vague and overall I'm not sure if this interpretation is well-supported:

WWTP effluent loads are typically constant, which would produce dilution-driven concentration variability (high CVc) but wouldn't necessarily indicate "dynamic" pressures because it would indicate consistent point-source loading being diluted by variable flow. Are the authors referring to other urban pressures? If so, the specific pressures are nebulous, which weakens the argument.

The wide distribution of $CV_C/CV_Q$ values within QMin-Synced (Figure S9) that overlaps with QMax-Synced suggests this class may contain distinct subgroups operating under different mechanisms.

High CVc/CVq can also occur in natural forested catchments (Ehrhardt et al. 2019). Given that woodland percentage appears higher in QMin than QMax catchments (Figure 5), could some of this variability reflect natural catchment dynamics rather than urban watershed dynamics?

Second, we tested whether the high CVc/CVq values in QMin-Synced catchments might be explained by agricultural legacy or forested dynamics, as suggested. Legacy-dominated agricultural systems generally exhibit weak or chemostatic C–Q behaviour and low temporal variability, because groundwater or soil-nitrogen stores release nitrate gradually over time (Winter et al., 2021). Johnson & Stets (2020) similarly show that legacy nitrate elevates winter low-flow concentrations but does not produce sharp annual peaks or strong flow sensitivity. These signatures differ from our QMin-Synced catchments, which in general exhibit steep negative C–Q slopes, high CVc/CVq, and a single annual concentration peak aligned with the minimum-flow month, patterns consistent with hydrological dilution of continuous urban inputs rather than gradual legacy drainage. Our correlation analyses further support this distinction: CVc/CVq increases strongly with urban land and population density ($\rho = 0.55$ and $0.52$, $p < 0.05$) and decreases with arable land ($\rho = -0.59$, $p < 0.05$) and woodland cover ($\rho = -0.61$, $p < 0.01$). Woodland cover is low (median = 8%, IQR=4%) across all QMin sites. Thus, all these results could suggest legacy nitrogen likely contributes to concentration levels in some QMin catchments but does not dominate the temporal variability or the synchrony mechanism. These additional analyses support that QMin-Synced catchments are heterogeneous, but this heterogeneity is best described as a gradient in urbanisation and land-use composition, and the urbanisation may remain the primary influencing impacts on the QMin-Synchrony behaviour.

We will add the following sentence in the Method 2.4: "To test the possible internal heterogeneity of QMin-Synced catchments, we also calculated Spearman correlations among land-use variables and two nitrate metrics (median concentrations and CVc/CVq)."

We will add more details in the Result 3.3: "For a further spearman correlation within QMin-synced catchments, median nitrate concentrations showed positive tendencies with arable land ($\rho = 0.41$, $p = 0.11$) and urban land ($\rho = 0.42$, $p = 0.10$), and a significant negative association with grassland cover ($\rho = -0.50$, $p = 0.04$). In contrast, CVc/CVq was most strongly correlated with urban land ($\rho = 0.55$, $p < 0.05$), population density ($\rho = 0.52$, $p < 0.05$), arable land ($\rho = -0.59$, $p < 0.05$) and woodland cover ($\rho = -0.61$, $p < 0.01$). All the other correlation results are shown in Fig. S6."

We will revise the sentences in the Discussion: "A legacy-nitrogen explanation for QMin-synchrony, whereby slower drainage of stored soil or groundwater nitrate could elevate concentrations during low-flow periods, would be consistent with the large literature on this topic (Johnson and Stets, 2020). Our Spearman correlations suggest that agricultural legacy may indeed contribute to background nitrate levels in some QMin-Synced catchments. Legacy-dominated agricultural systems typically show weak or chemostatic C–Q behaviour and comparatively low temporal variability because nitrate is released gradually from subsurface stores (Winter et al., 2021).

At the same time, several characteristics of our QMin-Synced catchments point to a stronger influence of continuous urban inputs on the synchrony pattern. These catchments feature steeply negative C–Q slopes, consistent with hydrological dilution of relatively stable urban point sources. Moreover, if both diffuse and point sources were active, we would likely expect dual peaks, one during winter flushing (as with QMax-Synched catchments) and another at low flow, yet only a single low-flow maximum is observed. This pattern further implies that urban

land and population density associated inputs have largely displaced the diffuse, winter-mobilisation behaviour typical of QMax-Synched catchments, creating an engineered inversion where nitrate concentrations peak only under low-flow conditions and are otherwise easily diluted (Kaushal and Belt, 2012; Kaushal et al., 2011). Our correlation analyses further support the interpretation that variability (CVc/CVq) increases with urbanisation, whereas arable land is associated with reduced variability, suggesting a more stable, weakly flow-responsive behaviour typical of legacy-influenced systems. Together, while legacy nitrate contributes to background concentration levels in some QMin-Synced catchments, the observed QMin-synchrony may be shaped primarily by flow-dependent dilution of continuous urban inputs."

[Figure]

**Figure S6: Spearman Correlation matrix for land-use variables and nitrate metrics in QMin-Synced catchments; Colours indicate correlation strength and direction. Asterisks denote significance: p < 0.05 (\*), p < 0.01 (\*\*), p < 0.001 (\*\*\*).**

**Specific comments**

- Section 2.2: WRTDS is a complex model, and given that the authors use its model parameters directly, the manuscript would benefit from subtle but important clarification on the model methods. For instance, "fitted through regression at each time point" is ambiguous. The readers might not understand that coefficients are estimated for each modeled day, meaning each station has thousands of $\beta_2$ values. Additionally, there is no mention of the time, season, and discharge window widths used, which affect how rapidly the estimated C-Q relationship can change over time. Did you choose default window sizes or select each catchment window them based on model fit?

We agree and thank the reviewer for highlighting this important point. We will clarify the WRTDS fitting procedure in Sect. 2.2. Specifically, we now explain that WRTDS performs a

locally weighted regression for each modelled day, such that every catchment has thousands of sets of coefficients that vary smoothly over time. We will explicitly report the window widths governing the local weighting. In this study, we used the default EGRET settings, namely half-window widths of 7 years in time, 0.5 years in season, and 2 natural-log units in discharge. These values are the recommended defaults in EGRET and are widely applied in WRTDS studies, ensuring methodological consistency across all catchments.

The revised text will clarify and resolve the ambiguity noted by the reviewer: "The WRTDS method estimates a smooth concentration surface in time–discharge–season space by performing a locally weighted regression for every modelled day, producing thousands of smoothly varying coefficient vectors for each catchment rather than a single global fit. The local weighting is controlled by three half-window widths: 7 years in time, 0.5 years in season, and 2 natural-log units in discharge."

- Line 169-172: Confidence in the models could be strengthened through more detailed model assessment. While observed vs. modeled concentration plots are provided in supplementary material, the main text should clarify whether any sites were excluded based on fit criteria and whether poor model performance at certain sites affects interpretation of results. Additionally, RMSE is difficult to evaluate without knowing mean concentrations at each site. Using a normalized metric such as KGE, NSE, or PBIAS would aid interpretation in the manuscript.

We agree and thank the reviewer for this helpful suggestion. Following the other reviewer's recommendation, we will use the Spearman temporal correlation as a metric to assess whether WRTDS captures the temporal pattern relevant for synchrony detection. Because our analysis focuses on the timing of monthly concentration peaks, temporal correlations could provide a more appropriate assessment.

We will revise the results section 3.1 as follows: "To assess the temporal consistency, spearman correlations between observed and reconstructed concentrations were calculated showing a median ρ of 0.72 (IQR = 0.134). No sites were excluded based on model performance."

- Line 117: Was distinct unimodal pattern determined visually? Did any sites not have unimodal pattern at all sites and were therefore removed from the dataset? Or were these part of the Asynced class?

We agree and thank the reviewer for raising this question. The statement regarding a "distinct unimodal pattern" refers to the dominant seasonal cycle that characterises temperate UK catchments. Prior to the synchrony analysis, we verified that all sites exhibited a single, well-defined seasonal peak–trough structure based on their long-term monthly means and that annual maxima and minima in discharge and nitrate concentration could be consistently identified.

We will add the following clarification: "We confirmed that both discharge and nitrate concentration exhibit a clear seasonal peak–trough structure at all sites, based on their long-term monthly means."

- Section 2.4: Clarification on the RF model would strengthen the analysis. First, was there any hyperparameter tuning done or were default parameters used? Second, it is unclear whether the permutation importance rankings were averaged across the k-fold cross validation, or whether k-fold cross validation was used only to assess model performance and a single final model fit to all data used to generate permutation importance rankings.

We agree and thank the reviewer for this useful comment. We did not perform hyperparameter

tuning. Apart from setting num.trees = 500, all parameters followed the ranger defaults. The repeated 10-fold cross-validation was used solely to evaluate model performance and stability. Permutation importance was not averaged across cross-validation folds. Consistent with the variable-importance framework of Altmann et al. (2010), CV was used solely to assess classification performance, after which a final RF model was trained on the full dataset, and computed permutation importance and permutation-based p-values from this final model, using 1000 label permutations to generate the null distribution.

We will clarify this as follows: "To understand the catchment controls on the two synchrony patterns, a series of catchment descriptors (Table 1) were selected, and a Random Forest (RF) model was applied to relate synchrony type (QMax-Synced vs QMin-Synced) to catchment characteristics. RF analysis was chosen for its robustness and ability to handle complex interactions within the data (Breiman, 2001). Asynced catchments were excluded as this group represents catchments with no dominant synchrony pattern and high variability, which would reduce the clarity and interpretability of the results. A Random Forest classifier was implemented using the ranger (Version 0.16.0) engine via mlr3 (Version 0.22.1) in R (Lang et al., 2019; Wright and Ziegler, 2017). No hyperparameter tuning was performed; ranger default settings were used except that the number of trees was set to 500. A three-repeated 10-fold cross-validation procedure was solely used to evaluate model performance and its generalizability. Following cross-validation, a final RF model was trained on the full dataset, and permutation-based variable importance values were extracted from the underlying ranger model, following Altmann et al. (2010). Empirical permutation p-values (<0.05) were obtained from 1000 random permutations of the response variable to identify descriptors whose importance exceeded the null distribution."

- Section 3.4: This section is challenging to understand. In general, ternary plots are challenging to parse, and the current presentation does not provide sufficient guidance for interpretation. I suggest either providing more detailed guidance in the text walking the reader more carefully through the interpretation, and/or supplementing the plots with simpler plots to isolate the key relationships. Additionally, Figure S11 caption is not clear and should be expanded upon to help readers understand.

We thank the reviewer for this helpful comment. We agree that ternary plots can be difficult to interpret without explicit guidance.

We refined the paragraph with the starting sentence: While Section 3.3 identified catchment characteristics that explain dominant synchrony types across space, here we examine (i) what controls interannual variability in synchrony within catchments, and (ii) how long-term variability in synchrony composition relate to catchment attributes.

And more explicit guidance related to interpretation of the ternary plots will be included: "The ternary plots (Fig. 6) summarise the long-term synchrony composition at each site by showing the relative proportion of QMax-, QMin- and Asynced years. In these diagrams, each vertex represents 100% dominance of one synchrony type; positions along the edges indicate mixtures of two types; and points near the centre represent a balanced mixture of all three. The colour scale applied to each point represents values for a selected catchment descriptor (e.g., arable land percentage, Drainage Path Slope), allowing the ternary diagram to visualise how synchrony composition varies along environmental gradients. These colour gradients are consistent with the spatial correlations shown in Fig. S7. No catchment sits at a single vertex; instead, most exhibited a mix of synchrony types, with Asynced catchments spanning a particularly wide range. No sites are located near both the QMax and QMin vertices simultaneously, indicating that switching occurs mainly between a dominant mode and Asynced behaviour rather than directly between the two synchronous modes."

The caption of Figure S7 will be "Figure S7: Heatmaps showing Spearman correlations between catchment descriptors and the proportion of years classified as QMax-Synced, QMin-Synced, or Asynced within three groups of catchments. Coloured cells indicate the direction and strength of correlation ($\rho$), with warm colours denoting positive and cool colours denoting negative correlations. Asterisks mark statistically significant correlations ($p < 0.05$)".

**Technical corrections**

- *QMax-Synched* and *QMax-Synced* are both used throughout the manuscript.

We thank the reviewer for raising this point. We will standardise the terminology throughout the manuscript and revise all occurrences to "QMax-Synced" for consistency.

- Line 84: Fix table and citation parentheses.

We thank the reviewer for pointing this out. The parentheses will be corrected and the citation will be added in the table.

- Line 84: Please make sure the catchment features used are outlined more clearly. For example, looking at the NRFA citation, I find two BFI variables but the manuscript does not state which one is being used.

We thank the reviewer for highlighting this ambiguity. The catchment descriptor used in our analysis is BFIHOST, the FEH soil-based Baseflow Index derived from the Hydrology Of Soil Types (HOST) classification. It is the variable provided within the NRFA FEH catchment descriptor dataset. Table 1 will be updated to reflect this.

- Line 94: citation missing for WWTP data

We thank the reviewer for this kind reminding. The citation for WWTP data will be added as: "Environment Agency: Consented Discharges to Controlled Waters with Conditions. [dataset], available at: https://www.data.gov.uk/dataset/55b8eaa8-60df-48a8-929a-060891b7a109/consented-discharges-to-controlled-waters-with-conditions1, last access: [10 Oct 2024], 2024"

- Line 112: Subscript 2 in model coefficient

We thank the reviewer for pointing this out. We will correct the formatting of the coefficient and ensured that the subscript "2" appears properly in all instances of $\beta_2$ throughout the manuscript.

- Line 113: missing period

We thank the reviewer for noting this typographical issue. The missing period at Line 113 will be added.

- Line 305: spelling error in figure caption "(d) Population density Density of WWTPs" should be Density of WWTPs

We thank the reviewer for pointing this out. The error in the caption of Figure 6d will be corrected to "Density of WWTPs."

- Line 354: "In our study, random forest analysis identified urban area as the strongest explanatory variable in these catchments." This sentence is misleading. The random forest ranked urban LU highest in variable importance for the classification. It the predictor helps most with prediction accuracy, but it does not provide explanatory

inference. The significant differences between synchrony classes were shown in the Wilcoxon results, not by the RF. I'd recommend revising the phrasing to reflect this.

We agree thank the reviewer for this important clarification. The RF analysis provides a ranking of predictors based on their contribution to classification accuracy but does not itself imply explanatory causation. The statistical differences between synchrony classes are instead supported by the Wilcoxon rank-sum tests.

We will revise the sentence as follows: "In our study, Random Forest analysis identified urban land cover as the most influential predictor for distinguishing QMin- from QMax-synced catchments, with Wilcoxon tests confirming significant differences between the two groups."

References:

Bowes, M. J., Jarvie, H. P., Naden, P. S., Old, G. H., Scarlett, P. M., Roberts, C., Armstrong, L. K., Harman, S. A., Wickham, H. D., and Collins, A. L.: Identifying priorities for nutrient mitigation using river concentration–flow relationships: The Thames basin, UK, J. Hydrol., 517, 1-12, 10.1016/j.jhydrol.2014.03.063, 2014.

Cooper, R. J., Warren, R. J., Clarke, S. J., and Hiscock, K. M.: Evaluating the impacts of contrasting sewage treatment methods on nutrient dynamics across the River Wensum catchment, UK, Sci. Total Environ., 804, 150146, 10.1016/j.scitotenv.2021.150146, 2022.

Johnson, H. M. and Stets, E. G.: Nitrate in Streams During Winter Low-Flow Conditions as an Indicator of Legacy Nitrate, Water Resour. Res., 56, 10.1029/2019wr026996, 2020.

Hiscock, K. M., Cooper, R. J., Lovett, A. A., and Sünnenberg, G.: Export Coefficient Modelling of Nutrient Neutrality to Protect Aquatic Habitats in the River Wensum Catchment, UK, Environments, 10, 10.3390/environments10100168, 2023.

Winter, C., Tarasova, L., Lutz, S. R., Musolff, A., Kumar, R., and Fleckenstein, J. H.: Explaining the Variability in High-Frequency Nitrate Export Patterns Using Long-Term Hydrological Event Classification, Water Resour. Res., 58, 10.1029/2021wr030938, 2022.